# Ozonation procedure for removal of mycotoxins in maize: A promising screening approach for improvement of food safety

Piotr Antos[1]*, Justyna Szulc[2], Tomasz Ruman[3], Maciej Balawejder[4], Krzysztof Tereszkiewicz[1], Bożena Kusz[1]

1 Department of Computer Engineering in Management, Rzeszow University of Technology, Rzeszów, Poland, 2 Department of Environmental Biotechnology, Lodz University of Technology, Łódź, Poland, 3 Department of Inorganic and Analytical Chemistry, Rzeszow University of Technology, Rzeszów, Poland, 4 College of Natural Sciences, Institute of Food Technology and Nutrition University of Rzeszow, Rzeszow, Poland

* p.antos@prz.edu.pl

**Data Availability Statement:** All relevant data are within the paper.

**Funding:** The author(s) received no specific funding for this work.

## Abstract

Mycotoxins are well known secondary metabolites of various fungi. They pose a significant threat to human and animal when present in food or feed. They can be responsible for losses in grain production and in livestock or human intoxication. In this study, several mycotoxins were detected in *Aspergillus fumigatus* contaminated maize kernels. The contaminated kernels were treated with gaseous ozone at a concentration of 500 and 3000 ppm for 1 hour. Depending on the specific compound, the contamination level was reduced by up to 100%. This screening research showed that a concentration of ozone as high as 3000 ppm could be sufficient to completely remove several toxic compounds from the maize matrix.

## Introduction

Maize is one of the most important crops in the world and at the same time an ideal habitat for the development of toxin-producing fungi. Their development on the surface and inside of the grain depends on grain moisture content, temperature and storage time [1]. Currently, several cases of the occurrence of mycotoxins in various food products were reported including: rye, barley, oats, maize, or other matrixes such as milk [2–4]. Mycotoxins have profound impacts on human health and livestock productivity. A number of these compounds have been proven to be carcinogenic, mutagenic, and genotoxic [5]. Different types of mycotoxins can be formed depending on the matrix and species responsible for its colonisation. *Aspergillus* species can be considered one of the most hazardous environmental contaminants, and their metabolites can have a negative impact on human and animal health [6]. For example, *Aspergillus* species are responsible for colonisation of various food products and cause their contamination with a number of mycotoxins such as aflatoxins (AFT), ochratoxin A (OTA), patulin (PAT), citrinin (CIT), aflatrem (AT), secalonic acids (SA), cyclopiazonic acid (CPA), terrein (TR), sterigmatocystin (ST) and gliotoxin (GT) [6]. Other mycotoxins can also be formed depending on the type of matrix. Contamination by *Fusarium* species can result in the formation of

**Competing interests:** The authors have declared that no competing interests exist.

trichothecenes mycotoxins. There are two types of trichothecenes that cause significant public health concerns due to their toxicity and occurrence, i.e. (Type A and B). The type A group includes T-2 toxin (T-2), HT-2 toxin (HT-2), neosolaniol (NEO) and diacetoxyscirpenol (DAS) and the type B group includes DON (Deoxynivalenol), NIV (Nivalenol) and 3- and 15-acetyldeoxynivalenol (15-ADON) [7–9]. Another group of mycotoxins that can be responsible for intoxication are enniatins (ENN) [10].

The presence of these harmful substances in maize products has led to urgent research efforts to develop efficient and safe methods for their mitigation. Therefore, to address the problem of their occurrence in food products, a broader context of food safety regulations and consumer preferences is required. It consists of pre- or post-harvest solutions, including prevention and decontamination. In the article by Pankaj et al. (2018) [5], the authors described the general approach to solving the problem of mycotoxins in food products. Fungal contamination can be dealt with applying fungicides in the field or, later, during the applications of a wide range of methods during storage.

A promising path is based on food and feed product decontamination procedures. There are a number of physical, chemical, and biological paths to deal with mycotoxins. One of them is heating, but it requires a temperature range of 237 to 306°C to reduce aflatoxins [5]. Another physical approach is based on gamma irradiation, which was evidenced to cause a reduction in mycotoxins in food matrix [11].

Among chemical decontamination methods, the use of ozone seems to be the most promising approach, since this oxidative agent not only is very potent, but is also considered safe for use in combination with food products [12]. Ozonation is a procedure that involves utilization of an allotrope form of oxygen; ozone ($O_3$), to eliminate by the oxidation pathways various biological and chemical pollutants [13, 14]. In particular, its potential to degrade mycotoxins presents a valuable opportunity to address the challenge of AFT or DON contamination in maize or wheat, ensuring the achievement of both food safety and the improved economic viability of agricultural products [15, 16]. In 1997 [17] have already paved the way for future investigations, as they utilised ozone in order to degrade mycotoxins. However, they used artificially contaminated maizemeal (i.e., in a form of powder or solution), and investigated only the reduction of aflatoxins. Pankaj et al., [5] demonstrated that utilization of gaseous ozone at a concentration between 40 and 90 ppm allowed up to 88% reduction of aflatoxin during 5–40 minutes of exposure. In a more recent study [18], gaseous ozone at the concentration of 0.15 ppm was utilised for the reduction of aflatoxins in maize. The authors achieved 92.4% reduction at the exposure time of maize at 30 and 60 min. Other authors reported a reduction of more than 60% of ZEN in maize flour during exposure of the matrix to ozone at a concentration of 51 ppm for 20 to 60 minutes [19]. In [20] it was reported that DON, ZEN, and AFT levels in ozonated ground maize were reduced below the detection level after exposure to gaseous ozone and no oxidation byproducts were detected. The absence of oxidation byproducts in their research is important because in some cases, metabolites of toxic compounds remain toxic or become even greater threat to human and animal lives. In their research, they utilised 40, 70 and 85 ppm of ozone for 180 minutes.

In this study, our objective was to verify the efficacy of ozone treatment to remove a wide range of mycotoxins. Our goal was to obtain a maximal reduction of the fungal metabolites with a complete degradation without the generation of degradation products. As a result of these efforts, we obtained a 100% reduction for most of the detected compounds, at an exposition time of 60 min, and the ozone concentration was 500 or 1000 ppm. We cultivated selected fungi strain of *A. fumigatus* in order to achieve contamination with mycotoxins produced by real fungi instead of artificial spiking maize kernels in mycotoxin or utilizing ozone to degrade mycotoxin present in a liquid solution. We obtained a total reduction of the wide range of

mycotoxins detected, including: Beauvericin, Enniatins (B and B1), Deepoxy-deoxynivalenol, Ochratoxin A, HT2-toxin, without the presence metabolites of mycotoxin oxidation. Moreover, among the degraded toxic compounds, there were enniatins (ENNs) whose degradation has not been investigated before.

## Materials and methods

### Maize grain

For the research authors own grown maize grain was used. The maize grain used in the experiment came from a crop located in Poland, Podkarpackie Voivodeship, Jaroslaw County. Geographical coordinates: (50.0038885"N 22.9759221"E). The maize variety used was DKC 3088, FAO 230, and sowing was carried out around 22 April 2022. The following treatments were applied before sowing: urea addition with urease inhibitor (220 kg/ha), Omya Calciprill lime (granular chalk) (500 kg/ha), NPK Ultra 8-20-30 (300 kg/ha). Sprays applied after sowing: Adengo 315 SC (0.35 l/ha). After about a month, another foliar spray of IKANOS 040 (1 l/ha) and Basfoliar 2.0 36 Extra (4 l/ha) was applied. Harvest (delayed) at the end of October. Yield about 5–5.5 t/ha at 17–18% field moisture. The maize was dried to 14% moisture and stored in 25 ton silos.

### Test strain

*A. fumigatus* was obtained from Collection of Pure Culture at the Institute of Fermentation Technology and Microbiology at Lodz University of Technology (strain no. LOCK CPC 0600). This strain was genetically identified (ITS gene sequences deposited in the National Center for Biotechnology Information GenBank database with no. KC456184) and had confirmed toxinogenicity in the studies carried out by [21, 22].

### Maize grain contamination

1 kg of corn grain in 100 mL aqueous glucose solution (glucose 38 g/L) was placed in glass bottles of 2 L. The grains were allowed to stand at room temperature for 2 days to swell. Then, the samples were autoclaved (15 min, 121˚C), cooled down, and inoculated with 50 mL of spores suspension.

Spore suspensions were collected from a culture of *A. fumigatus* cultivated on two plastic Petri dishes (diameter of 90 mm) with a PDA medium (Potato Dextrose Agar, BTL, Poland) using a sterile swab and transferred into 100 mL of sterile water (with 0.05% Tween 80). The number of spores was determined using a Thom cell chamber and confirmed by the culture method. Subsequently, the spore suspensions were diluted to $10^6$ spores / mL concentration. The maize grains cultures were incubated at temperature 25±2˚C for 14 days.

### Chemical compounds used during the determination of mycotoxins

All chemicals were of analytical reagent grade. Deionized water (18 MΩ·cm) was prepared at home. LC-MS-grade methanol and acetonitrile were purchased from Aldrich (Poland). Syringe filters (PTFA membrane, 0.2 μm-pore) were purchased from Aldrich.

### Apparatus

Mass spectrometry-liquid chromatography analyses were performed with Bruker Elute UHPLC system operated by Hystar 3.3 software and a ultrahigh resolution (60000+) mass spectrometer Bruker Impact II (Bruker Daltonik GmbH) ESI QTOF-MS equipped with Data Analysis 4.2 (Bruker Daltonik GmbH), TASQ (2022b) and Metaboscape (2022b).

## TargetScreener analysis

A Bruker UHPLC Column Intensity Solo (C18 silica, 1.8 μm particles, 100x2.1 mm) with a column guard was used for the TargetScreener measurements. Two types of mobile phases were used: A = water/methanol (99:1) with 5 mM ammonium formate and 0.01% formic acid, B = methanol with 5 mM ammonium formate and 0.01% formic acid (v/v). The autosampler was thermostated at temperature of 4˚C. A volume of 5 μL of the extract was loaded into the column at a flow rate of 200 μLmin$^{-1}$, using 4% B. For the TargetScreener bbCID measurements, the B percentage was changed with time as follows: 1 min—18.3% B, 2.5 min—50%, 14–16 min 99.9%, 16.1–20 min– 4%. The solvent flow was 200 μLmin$^{-1}$ within 0–1 min, and then gradually changed from 200 to 223 μLmin$^{-1}$ within 1–2.5 min and from 223 μLmin$^{-1}$ to 400 μLmin$^{-1}$ within 2.5–14 min, then from 400 μLmin$^{-1}$ to 480 μLmin$^{-1}$ within 14 to 19 min, and back to 200 μLmin$^{-1}$ within 19.1 to 20 min. The TargetScreener results were analysed with the TASQ 2022b software with a built-in database of more than four thousand environmentally and toxicologically important compounds. The database contains spectral data (MS, MSMS spectra) as well as retention times of compounds. Only compounds with the highest matching parameters (retention time, precursor, and fragments parameters) were mentioned in this work. All samples were measured in triplicates.

## Sample preparation

Maize kernels (average 4.2 g) were transferred into a 50 mL falcon tube to which 10 mL of methanol was added. The falcon tubes were placed in an ultrasonic bath for 1 hour and then incubated at room temperature for 72 hours. After this time, the extracts were transferred to round bottom flasks and the grains were washed with an additional portion of methanol (1 mL) that was added to the extract. Solvents were evaporated using a rotary evaporator and then in the SpeedVac vacuum concentrator (at approximately 0.9 mbar vacuum). The dried materials were resuspended in 80 μL of acetonitrile and placed in an ultrasonic bath for 30 seconds, mixed using a vortex shaker (30 s), and then centrifuged at 9800 xg for 5 minutes. Finally, the 10-fold diluted samples were transferred to HPLC vials.

## Ozonation procedure

Kernel samples that were infected with fungi originated from the earlier grown colony (section: maize grain contamination) were later exposed to ozone gas generated from ambient air using the, the TS 30 ozone generator (Ozone solution) providing ozone at a concentration of 500 and 3000 ppm that was controlled with the UV-106-MH ozone analyzer for 1 hour.

# Results and discussion

## Results of targeted and untargeted UHPLC-HRMS analysis

The targeted identification of toxic compounds was performed with the TargetScreener system (Bruker Daltonics) (materials and methods section) using the bbCID measurement method, which is based on tight tolerances matching of analyte retention time, precursor *m/z*, fragments *m/z*, precursor and fragments isotopic pattern and also precursor-to-fragment signals intensity ratios (Table 1). The results obtained in the process of identification of the toxic compound in the samples of maize kernels are summarized in Fig 1. The data include the c/c0 ratio, that is the ratio of mycotoxin concentration after ozonation to the mycotoxin concentration before the detoxication procedure.

From Fig 1 it can be observed that a number of detected toxic compounds were affected by ozone-mediated oxidation process. As described above, several reviews and research articles

**Table 1. Identification parameters of toxic compounds found in maize kernels samples with the positive ion mode TargetScreener bbCID method.**

| Compound name | m/z meas. | RT [min] | Expected diagnostic ions | Diagnostic ions found | S/N | Delta RT [min] | Delta m/z [mDa] |
|---|---|---|---|---|---|---|---|
| Beauvericin (NH$_4$) | 801.4414 | 12.80 | 3 | 3 | 69 | -0.10 | -0.09 |
| Enniatin B (NH$_4$) | 657.4422 | 12.35 | 4 | 4 | 148 | -0.14 | -0.43 |
| Enniatin B1 (NH$_4$) | 671.4572 | 12.67 | 4 | 4 | 73 | -0.17 | -1.79 |
| Mycophenolic acid | 351.1329 | 8.12 | 2 | 2 | 1810 | -0.03 | -0.38 |
| Deepoxy-deoxynivalenol | 281.1379 | 4.98 | 1 | 1 | 4361 | 0.33 | -0.48 |
| Ochratoxin A | 404.0888 | 8.44 | 2 | 2 | 56 | 0.05 | -0.75 |
| HT2-toxin (NH$_4$) | 442.2430 | 7.55 | 1 | 1 | 76 | -0.07 | -0.54 |

have been published on the topic of mycotoxins degradation in food products. Ozone has been reported to show a highly promising potential for aflatoxin degradation, e.g. in the review article by [5] the mechanism of mycotoxin degradation was briefly described. It was based on an electrophilic attack on double bond between C8-C9 of the furan ring, which resulted in the formation of ozonides followed by rearrangement into monozonide derivatives, e.g., aldehydes, ketones and organic acids. However, in our research, to completely degrade mycotoxins without a risk of generation of byproducts, which could still pose a threat to human and livestock health, we utilized gaseous ozone at the concentration ranged from 500 up to 3000 ppm (Fig 1). This was proven to be sufficient for 100% reduction of most of detected compounds, and no products of mycotoxin degradation were detected.

One of the compounds present in the infected grains was beauvericin. This compound is acyclic hexa depsipeptide that is synthesized by various fungi. It exhibits potent antimicrobial properties against a wide range of bacteria, affecting both Gram-positive and Gram-negative strains equally. Furthermore, this toxin also shows cytotoxic, apoptotic, and immunosuppressive effects. Beauvericin primarily targets cell membranes, increasing their permeability and disrupting cellular balance [8, 23]. The exact mechanism of its action is not entirely clear, but it is suspected to involve mitochondrial damage, similar to the mechanism seen with enniatins. We observed that during the exposition of this compound to ozone at a concentration of 500 ppm, a reduction between 10% and 15% of the initial amount of compound was achieved based on the peak area of the detected mycotoxins. At the 3000 ppm ozone concentration, the total reduction of the present mycotoxin was achieved (e.g., no signal of this compound was detected after exposure to ozone at 3000 ppm for 1 hour).

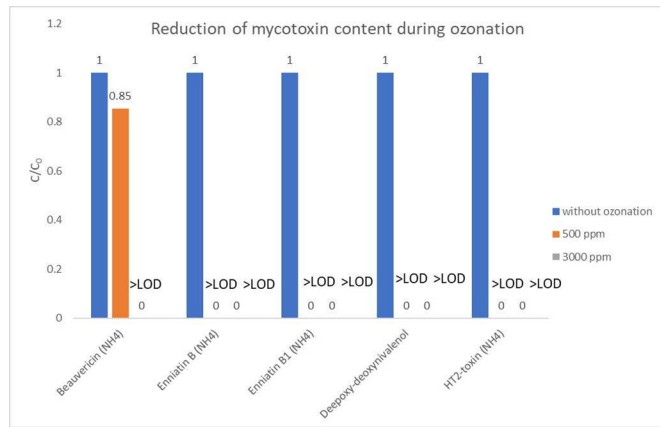

**Fig 1. Reduction of mycotoxin content in the maize matrix during exposure to gaseous ozone.**

Another group of toxic compounds generated by fungi were enniatin molecules (Fig 1), which contain a cyclohexadepsipeptide structure, featuring a repeating pattern of three N-methyl amino acids and three hydroxyl acids. They have a lipophilic nature, allowing them to embed into lipid layers of the cell membrane. Enniatins exhibit various properties, including antimicrobial, insecticidal and antifungal characteristics, and they may also have herbicidal effects. These compounds act by targeting cellular membrane transport proteins, inhibiting their function. Enniatins have a significant impact on mitochondria, essential organelles responsible for cellular respiration, and the production of adenosine triphosphate (ATP), which is vital for energy transfer in cells. Although ENN appears to be efficiently decomposed in the digestive systems of animals, further research is required in this regard [8]. In our study, this group of enniatin, the B and B1 types were present in kernels infected with fungi present in maize. The exposed samples revealed a 100% reduction without detection of secondary decomposition compounds. The literature is lacking reports on ENN molecules degradation with ozone.

Moreover, we detected a number of mycotoxins from a trichothecenes group in infected maize kernels. It was type A trichothecenes, for example, HT-2 toxin (HT-2), and type B trichothecenes, which included deoxynivalenol (DON). In [24] the reduction of DON in an acetonitrile solution subjected to ozonation was reported. The ozonation time of the acetonitrile solution was conducted for 9 min and some metabolites were detected. In their research, the authors were able to propose a possible course of transformation of DON into other compounds. In another study, it was observed that in the case of DON molecules, ozone attacks double bonds at C9-C10 in the DON molecule and also oxidises allylic carbon in the 8 position [12]. In our research, the reduction was 100% for both HT2 and DON, that is below the limit of the detection level and no metabolites were detected. Moreover, detoxication occurred in conditions similar to real life as the naturally contaminated matrix were maize kernels.

Last but not least, mycophenolic acid and ochratoxin A were detected in the maize kernels subjected to infection with the strain of used fungi. However, their concentration was low and showed no variation in concentration between control and ozonated samples.

In general, although the mechanism of detoxification of mycotoxin compounds mediated by ozone molecules is not entirely clear for a number of toxic compounds, still it can be expected that the oxidizing agent reacts with the functional groups in the mycotoxin molecules, therefore affecting their molecular structures. As a result, different products characterised by lower molecular weight and fewer double bonds and toxicity are formed [12]. Moreover, the optimal ozone exposure time and ozone concentration varies in case of different fungal species and matrixes. Therefore, for particular matrixes and particular mycotoxins occurring in them, a suitable research is needed [2]. Although there is still a literature gap on several aspects of maize ozonation, the investigated technology has already been commercialized by the Eye-Grain ApS company in a form of a large-scale processing line which is intended for ozonation of contaminated food commodities https://crop-protector.com/wp-content/uploads/2022/12/toxi-scrub-brochure-mail-distribution-en-0151-000c.pdf).

## Conclusions

Although there are a number of approaches that can lead to reduction of mycotoxin in food products, the ozonation procedure may be considered as an important tool with high potential, for improvement of food safety. We aimed to explore the utilization of ozonation as a suitable method for the oxidation of various naturally produced mycotoxins in maize kernels intended to feed livestock. Although there are some reports on the reduction of mycotoxin by ozone, reports including a real-life reflecting matrix and storage conditions (i.e. ozonation of

grain in silos) are scare. In this research, we evidenced that several toxic compounds belonging to mycotoxins could be reduced during exposition of maize kernels infected with fungi to ozone gas at a concentration of 500 or 3000 ppm for no longer than 1 hour. Furthermore, as a result of mycotoxin degradation, no metabolite products were detected, indicating total mineralization. Further research will aim at reducing the concentration of ozone in order to find the lowest effective concentration and utilization of other matrixes and strains of fungi. The manuscript emphasizes the need for innovative and efficient mycotoxin decontamination techniques and presents ozonation as a promising solution for enhancing the safety and quality of maize products. This study may contribute to the growing body of knowledge in the field of food safety.

## Author Contributions

**Conceptualization:** Piotr Antos.

**Formal analysis:** Piotr Antos.

**Investigation:** Piotr Antos, Tomasz Ruman.

**Methodology:** Piotr Antos, Justyna Szulc, Tomasz Ruman, Maciej Balawejder.

**Resources:** Justyna Szulc, Tomasz Ruman, Krzysztof Tereszkiewicz, Bożena Kusz.

**Software:** Tomasz Ruman.

**Supervision:** Piotr Antos.

**Visualization:** Tomasz Ruman, Maciej Balawejder.

**Writing – original draft:** Piotr Antos.

**Writing – review & editing:** Piotr Antos.

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
