## [Decision Letter · Decision Letter 0]

24 Jun 2024

PONE-D-24-16519Ozonation procedure for removal of mycotoxins in maize: a Promising screening approach for improvement of food safetyPLOS ONE

Dear Dr. Antos,

Thank you for submitting your manuscript to PLOS ONE. After careful consideration, we feel that it has merit but does not fully meet PLOS ONE’s publication criteria as it currently stands. Therefore, we invite you to submit a revised version of the manuscript that addresses the points raised during the review process.

We look forward to receiving your revised manuscript.

Kind regards,

Amitava Mukherjee, ME, Ph.D.

Academic Editor

PLOS ONE

Journal Requirements:

Reviewers' comments:

Reviewer's Responses to Questions

**Comments to the Author**

1. Is the manuscript technically sound, and do the data support the conclusions?

Reviewer #1: Yes

Reviewer #2: Partly

2. Has the statistical analysis been performed appropriately and rigorously? 

Reviewer #1: N/A

Reviewer #2: No

3. Have the authors made all data underlying the findings in their manuscript fully available?

Reviewer #1: Yes

Reviewer #2: Yes

4. Is the manuscript presented in an intelligible fashion and written in standard English?

Reviewer #1: Yes

Reviewer #2: No

5. Review Comments to the Author

Reviewer #1: The manuscript submitted for review (ID PONE-D-24-16519) contains very interesting data that may have practical applications. In general, the work is relatively well written (minor comments below) and its individual chapters are coherent. Although the results are interesting, the scope of the research and the lack of use of a reference fungus strain from the world's collection of microorganisms make me very concerned about these studies. Therefore, I believe that the work is preliminary research that can be published rather as a scientific communication, of course after the authors respond to my suggestions below.

My main comments:

1) 2.2 Test strain: I understand that the tested strain was from the collection of a facility (scientific unit) that was identified by sequencing the ITS rDNA regions. What about the control strain? Why wasn't a control strain from some global culture collection tested e.g. ATCC?

2) Pg 4 ln 127: “Petri dish” - what size? what is it made of (plastic / glass)?

3) Fig. 1: please write numbers with a dot

My little suggestions:

1) Pg 1 Ln 34: remove dot „mutagenic and genotoxic. (Pankaj et”

2) Pg 3 Ln 93: please use the abbreviation of the fungal name - Aspergillus fumigatus

3) Pg 4 ln 106: “Jarosław district” – pleas add “Poland” and geographical coordinates

4) Pg 4 ln 127: please use italics for A. fumigates

5) Pg 7 ln 209: remove dot

Reviewer #2: The manuscript reports the influence of ozonation on deactivation of mycotoxins in maize grains artificially colonized by Aspergillus fumigatus. All the five mycotoxins studied were almost completely deactivated by ozone treatment. The information generated by the authors are new and significant.

Despite the novel information generated, the presentation of the research needs significant improvement. Several experimental details and statistical treatment of experiments are missing. Some parts of the paper are unclear for the readers to understand or repeat the work. The authors need to edit the paper and make it more readable

A few comments are as follows:

• Line 30: Fungal development also depends on temperature. Add this factor in the sentence.

• L31: Replace ‘a number of’ with ‘several’

• L67-90: Require editing to make the text more understandable and accurate.

• Introduction: Please mention the objectives of the study and its significance in the final paragraph. The authors mention what they did and the summary results in the final paragraph of the Introduction, but not the objectives.

• L105-113: Needs editing. Disjointed incomplete sentences. Improve.

• L127: Italicize A. fumigatus

• L131: 106 spores/mL --- 6 should be superscript

• L157: Replace ‘four thousands of’ with ‘four thousand’

• L164-165: Replace ’A weighed amount of maize kernels (average 4.2 g) was transferred into a 50 mL falcon tube. Then, 10 mL of methanol was added’ with ‘Maize kernels (average 4.2 g) were transferred into a 50 mL falcon tube to which 10 mL of methanol was added’.

• L164: Clarify if the maize grains were colonized or not. If the maize grains were colonized, switch sections 2.7 and 2.8.

• L176: Provide details of ozone generator.

• Materials and Methods: Which experimental design was used? How many replicates/repetitions were used?

• Discussion appears to be disjointed with several very short paragraphs. The authors should also review information on ozonation for aflatoxin deactivation, which has reached commercial scale.

• Fig 1: Explain Y-axis (C/C0)

6. PLOS authors have the option to publish the peer review history of their article (what does this mean?). If published, this will include your full peer review and any attached files.

Reviewer #1: **Yes: **prof. dr Rafał Ogórek

Reviewer #2: **Yes: **Ranajit Bandyopadhyay

---

## [Author Response · Author response to Decision Letter 0]

9 Aug 2024

We thank the reviewers for their reviews, which allowed us to improve the quality of the paper.

Reviewer #1: The manuscript submitted for review (ID PONE-D-24-16519) contains very interesting data that may have practical applications. In general, the work is relatively well written (minor comments below) and its individual chapters are coherent. Although the results are interesting, the scope of the research and the lack of use of a reference fungus strain from the world's collection of microorganisms make me very concerned about these studies. Therefore, I believe that the work is preliminary research that can be published rather as a scientific communication, of course after the authors respond to my suggestions below.

My main comments:

1) 2.2 Test strain: I understand that the tested strain was from the collection of a facility (scientific unit) that was identified by sequencing the ITS rDNA regions. What about the control strain? Why wasn't a control strain from some global culture collection tested e.g. ATCC?

Answer:

The reviewer is right. We used the Aspergillus fumigatus LOCK CPC 0600 strain from the Collection of Pure Culture at Lodz University of Technology. This strain was genetically identified based on the ITS gene sequences (accession number in GenBank KC456184). 

We chose this strain because we had to be sure that it would produce mycotoxins; otherwise there would be no point testing mycotoxin reduction using the tested method.

In the case of this strain, we confirmed mycotoxin production in previous studies (Gutarowska et al., 2014; Szulc and Ruman, 2020). Of course, we are aware that collected strains should also be included in further research and we have prepared a scientific project with such research planned, which is currently being assessed. We aim to continue this research on strains purchased from the American Type Culture Collection (ATCC).

References:

Gutarowska B, Skóra J, Stępień Ł, Twarużek M, Błajet-Kosicka M, Otlewska A, Grajewski J. Estimation of fungal contamination and mycotoxin production at workplaces in composting plants, tanneries, archives and libraries, World Mycotoxin Journal, 2014, 7(3), 345–355; https://doi.org/10.3920/WMJ2013.1640

Szulc J, Ruman T. Laser Ablation Remote-Electrospray Ionisation Mass Spectrometry (LARESI MSI) Imaging—New Method for Detection and Spatial Localization of Metabolites and Mycotoxins Produced by Moulds. Toxins 2020, 12, 720. https://doi.org/10.3390/toxins12110720

2) Pg 4 ln 127: “Petri dish” - what size? what is it made of (plastic / glass)? 

We would like to clarify to the Reviewer that the molds were grown on PDA medium (Potato Dextrose Agar, BTL, Poland) in two plastic Petri dishes with a diameter of 90 mm from which the spores were collected using sterile swab sterile water with Tween 80. 

According to the reviewer's comment, we have added such details in the methodology of our research as follows:

Action taken: 

Section “Maize grain contamination” lines 133-137 were edited: 'Spore suspensions were collected from a culture of A. fumigatus on two plastic Petri dishes (diameter of 90 mm) with a PDA medium (Potato Dextrose Agar, BTL, Poland) using a sterile swab and transferred to 100 ml of sterile water (with 0.05% Tween 80). The number of spores was determined using a Thom cell chamber and confirmed by the culture method”.

3) Fig. 1: please write numbers with a dot

Action taken: 

Figure 1 was edited and “,” were replaced with “.”

My little suggestions:

1) Pg 1 Ln 34: remove dot „mutagenic and genotoxic. (Pankaj et”

Action taken: 

Section Introduction line 39 (dot removed)

2) Pg 3 Ln 93: please use the abbreviation of the fungal name - Aspergillus fumigatus

Action taken: 

Section “Introduction” line 98 Aspergillus abbreviated

3) Pg 4 ln 106: “Jarosław district” – pleas add “Poland” and geographical coordinates

Action taken: 

lines 109-118 were edited 

4) Pg 4 ln 127: please use italics for A. fumigates

Action taken: 

Section “Maize grain contamination” Line 133 italics used.

5) Pg 7 ln 209: remove dot

Action taken: done

Reviewer #2: The manuscript reports the influence of ozonation on deactivation of mycotoxins in maize grains artificially colonized by Aspergillus fumigatus. All the five mycotoxins studied were almost completely deactivated by ozone treatment. The information generated by the authors are new and significant.

Despite the novel information generated, the presentation of the research needs significant improvement. Several experimental details and statistical treatment of experiments are missing. Some parts of the paper are unclear for the readers to understand or repeat the work. The authors need to edit the paper and make it more readable

A few comments are as follows:

• Line 30: Fungal development also depends on temperature. Add this factor in the sentence.

Action taken: 

Section “Introduction”, line 34, 'temperature' was added 

• L31: Replace ‘a number of’ with ‘several’

Action taken: 

Section “Introduction”, line 35, “A number of” was replaced with “several”

• L67-90: Require editing to make the text more understandable and accurate.

Action taken: 

Section “Introduction”, lines 71-95, the text was edited. 

• Introduction: Please mention the objectives of the study and its significance in the final paragraph. The authors mention what they did and the summary results in the final paragraph of the Introduction, but not the objectives.

Action taken: 

Section “Introduction”, lines, 92-94 Introduction section was improved by addition of “In this study, our objective was to verify the efficacy of ozone treatment to remove a wide range of mycotoxins. The authors' goal was to obtain a maximal reduction of the fungal metabolites with a complete degradation without the generation of degradation products.

• L105-113: Needs editing. Disjointed incomplete sentences. Improve.

Action taken: 

Section “ Maize grain” lines 109-118, the mentioned paragraph was rewritten. 

• L127: Italicize A. fumigatus

Action taken:

Sections “Test strain” and “Maize grain contamination”, lines 121 and 133, italics was used.

• L131: 106 spores/mL --- 6 should be superscript

Action taken: 

Section „Maize grain contamination” line 137 was edited

• L157: Replace ‘four thousands of’ with ‘four thousand’

Action taken: 

Section “TargetScreener analysis” line 164 was edited

• L164-165: Replace ’A weighed amount of maize kernels (average 4.2 g) was transferred into a 50 mL falcon tube. Then, 10 mL of methanol was added’ with ‘Maize kernels (average 4.2 g) were transferred into a 50 mL falcon tube to which 10 mL of methanol was added’.

Action taken: 

Section “Sample preparation” line 171-172, maize kernels (average 4.2 g) were transferred into a 50 mL falcon tube to which 10 mL of methanol was added. replaced earlier sentence.

• L164: Clarify if the maize grains were colonized or not. If the maize grains were colonized, switch sections 2.7 and 2.8.

Action taken: 

Section “Ozonation procedure” lines 183-184, the chronological description of actions in the M&M section was improved.

• L176: Provide details of ozone generator.

Action taken: 

Section “Ozonation procedure” line 185, A TS 30 ozone generator (Ozone solution) was added.

• Materials and Methods: Which experimental design was used? How many replicates/repetitions were used? 

Action taken: 

Section “TargetScreener analysis” line 168, 'All samples were measured in triplicates.' was added

• Discussion appears to be disjointed with several very short paragraphs. The authors should also review information on ozonation for aflatoxin deactivation, which has reached commercial scale.

Action taken: 

Section “Results and Discussion” Authors joined some paragraphs and applied some corrections. Also Lines 265-273 were added.

• Fig 1: Explain Y-axis (C/C0) 

Action taken: 

Section “Results and Discussion” lines 194-196, a brief description “The data include the c/c0 ratio, that is the ratio of mycotoxin concentration after ozonation to the mycotoxin concentration before the detoxication procedure..” was added.

---

## [Decision Letter · Decision Letter 1]

29 Aug 2024

Ozonation procedure for removal of mycotoxins in maize: a promising screening approach for improvement of food safety

PONE-D-24-16519R1

Dear Dr. Antos,

We’re pleased to inform you that your manuscript has been judged scientifically suitable for publication and will be formally accepted for publication once it meets all outstanding technical requirements.

Kind regards,

Amitava Mukherjee, ME, Ph.D.

Academic Editor

PLOS ONE

Additional Editor Comments (optional):

Reviewers' comments:

Reviewer's Responses to Questions

**Comments to the Author**

1. If the authors have adequately addressed your comments raised in a previous round of review and you feel that this manuscript is now acceptable for publication, you may indicate that here to bypass the “Comments to the Author” section, enter your conflict of interest statement in the “Confidential to Editor” section, and submit your "Accept" recommendation.

Reviewer #1: All comments have been addressed

Reviewer #2: (No Response)

2. Is the manuscript technically sound, and do the data support the conclusions?

Reviewer #1: Yes

Reviewer #2: Yes

3. Has the statistical analysis been performed appropriately and rigorously? 

Reviewer #1: Yes

Reviewer #2: Yes

4. Have the authors made all data underlying the findings in their manuscript fully available?

Reviewer #1: Yes

Reviewer #2: Yes

5. Is the manuscript presented in an intelligible fashion and written in standard English?

Reviewer #1: Yes

Reviewer #2: Yes

6. Review Comments to the Author

Reviewer #1: The current version of the manuscript is better than the original. The authors took into account my suggestions regarding the work. My congratulations.

Reviewer #2: I cannot assess if some of the revisions in the manuscript were made or not since the revisions made based on the comments made during the first review is difficult to follow from the author responses. Line numbers mention in the Action Taken part of the response does not coincide with the text. A few examples of the discrepancies are: L164-165: Replace ’; L164: Clarify if the;

7. PLOS authors have the option to publish the peer review history of their article (what does this mean?). If published, this will include your full peer review and any attached files.

Reviewer #1: **Yes: **Rafał Ogórek

Reviewer #2: **Yes: **Ranajit Bandyopadhyay

---

## [Editor Report · Acceptance letter]

1 Sep 2024

PONE-D-24-16519R1 

PLOS ONE

Dear Dr. Antos, 

I'm pleased to inform you that your manuscript has been deemed suitable for publication in PLOS ONE. Congratulations! Your manuscript is now being handed over to our production team.

Kind regards, 

on behalf of

Professor Dr. Amitava Mukherjee 

Academic Editor

PLOS ONE